# TGF-β Modulated Pathways in Colorectal Cancer: New Potential Therapeutic Opportunities

**DOI:** 10.3390/ijms25137400

**Published:** 2024-07-05

**Authors:** Morena Fasano, Mario Pirozzi, Chiara Carmen Miceli, Mariateresa Cocule, Michele Caraglia, Mariarosaria Boccellino, Pasquale Vitale, Vincenzo De Falco, Stefano Farese, Alessia Zotta, Fortunato Ciardiello, Raffaele Addeo

**Affiliations:** 1Division of Medical Oncology, Department of Precision Medicine, University of Campania Luigi Vanvitelli, 80138 Naples, Italy; morena.fasano@unicampania.it (M.F.); mario.pirozzi@unicampania.it (M.P.); chiaracarmenmiceli@gmail.com (C.C.M.); mariateresacocule1292@gmail.com (M.C.); stefano.farese93@gmail.com (S.F.); alessiazotta@libero.it (A.Z.); fortunato.ciardiello@unicampania.it (F.C.); 2Department of Precision Medicine, University of Campania “L. Vanvitelli”, 80138 Naples, Italy; michele.caraglia@unicampania.it; 3Laboratory of Precision and Molecular Oncology, Biogem Scarl, Institute of Genetic Research, Contrada Camporeale, 83031 Ariano Irpino, Italy; 4Oncology Operative Unit, Hospital of Frattamaggiore, ASLNA2NORD, Frattamaggiore, 80027 Naples, Italy; pasquale.vitale3@aslnapoli2nord.it (P.V.); vincenzo.defalco1@aslnapoli2nord.it (V.D.F.); raffaele.addeo@aslnapoli2nord.it (R.A.)

**Keywords:** angiogenesis, colorectal cancer, EMT, microenvironment, TGF-β

## Abstract

Colorectal cancer (CRC) is the third most commonly diagnosed cancer worldwide, with 20% of patients presenting with metastatic disease at diagnosis. TGF-β signaling plays a crucial role in various cellular processes, including growth, differentiation, apoptosis, epithelial-mesenchymal transition (EMT), regulation of the extracellular matrix, angiogenesis, and immune responses. TGF-β signals through SMAD proteins, which are intracellular molecules that transmit TGF-β signals from the cell membrane to the nucleus. Alterations in the TGF-β pathway and mutations in SMAD proteins are common in metastatic CRC (mCRC), making them critical factors in CRC tumorigenesis. This review first analyzes normal TGF-β signaling and then investigates its role in CRC pathogenesis, highlighting the mechanisms through which TGF-β influences metastasis development. TGF-β promotes neoangiogenesis via VEGF overexpression, pericyte differentiation, and other mechanisms. Additionally, TGF-β affects various elements of the tumor microenvironment, including T cells, fibroblasts, and macrophages, promoting immunosuppression and metastasis. Given its strategic role in multiple processes, we explored different strategies to target TGF-β in mCRC patients, aiming to identify new therapeutic options.

## 1. Introduction

Colorectal cancer (CRC) is the third most commonly diagnosed malignancy worldwide and ranks second in cancer-related mortality [1,2]. Its incidence is 25% higher in males and differs greatly between countries. In the European Union, CRC was estimated to account for 12.7% of all new cancer diagnoses and 12.4% of all cancer-related deaths in 2020 [3].

In patients with CRC, 5-year survival varies from 28.5% to 57% in men and 30.9% to 60% in women, depending on the stage of the disease at the moment of diagnosis. The 5-year survival drops to <10% for patients diagnosed at stage IV [2]. At the time of diagnosis, 20% of patients present with metastatic disease [4], mainly liver, peritoneum, and lung but also brain and bone [5].

Although there are new surgical approaches and loco-regional treatments for liver and lung metastases, the prognosis of patients with metastatic CRC (mCRC) remains poor. Therefore, it is required to identify the underlying mechanisms of cancer development and metastatization in order to optimize the therapeutic strategies [6,7].

Transforming growth factor-beta (TGF-β) signaling is one of the most important pathways playing critical roles in many biological processes, including cell growth, differentiation, proliferation, apoptosis, epithelial–mesenchymal transition (EMT), extracellular matrix (ECM) remodeling, angiogenesis, and cellular immune responses and homeostasis [6]. The TGF-β superfamily contains more than forty members, including TGF-βs, Nodal, Activin, and bone morphogenetic proteins (BMPs). In the early stages of tumorigenesis TGF-β protein has a suppressive role by inducing cell cycle arrest and apoptosis in the early stages of tumor formation. During cancer progression, tumor cells gradually become resistant and secrete TGF-β themselves [8], working, in turn, as immunosuppressors facilitating neo-angiogenesis and tumor invasion and metastasis [9].

TGF-β ligands are regulatory cytokines that play an important role in different tumorigenic processes [6,8]. Alterations of TGF-β signaling can determine the development of a variety of tumors, including esophageal cancer, hepatocellular, pancreatic, gastric, and colorectal cancer [10,11,12].

In the present manuscript, we analyzed TGF-β signaling and investigated the role of TGF-β-dependent pathway in CRC pathogenesis. Moreover, we have described the mechanisms by which TGF-β influences metastases development, recognizing the involved key components. We also looked into potential clinical applications as therapeutic targets.

## 2. TGF-β Signaling Pathway

TGF-β superfamily signaling involves 30 components, categorized into two main subfamilies: the TGF-β-activin-nodal subfamily and the BMP subfamily [6,13,14]. There are three isoforms of TGF-β, TGF-β1, TGF-β2, and TGF-β3, each interacting with specific surface receptors known as TGF-βR1, TGF-βR2, and TGF-βR3 [14]. TGF-β ligands are secreted in an inactive form within the extracellular matrix and can be activated by integrins αvβ6 or αvβ8 [15,16].

Upon activation, TGF-β ligands bind to TGF-βR2, prompting the formation of a hetero-tetrameric complex with TGF-βR1 [10,17]. This complex phosphorylates TGF-βR1, which then phosphorylates the SMAD2 and SMAD3 proteins. SMAD proteins are central mediators of the TGF-β signaling pathway and are divided into three categories: receptor-regulated SMADs (R-SMADs, like SMAD2 and SMAD3), inhibitory SMADs (I-SMADs), and common-mediator SMADs (Co-SMADs, like SMAD4) [18].

Once phosphorylated, R-SMADs (SMAD2 and SMAD3) form a complex with SMAD4, which translocates to the nucleus to regulate gene expression [19,20]. In the nucleus, they influence various cellular processes, such as proliferation, differentiation, and apoptosis. After engaging in gene transcription, the linker region of R-SMADs is phosphorylated by cyclin-dependent kinase 8 (CDK8). Subsequently, glycogen synthase kinase 3 (GSK3) phosphorylates R-SMADs, creating a binding site for SMAD ubiquitination regulatory factor 1 (SMURF1) and other E3 ubiquitin ligases, leading to their ubiquitination and degradation by the proteasome [15].

Mutations in TGF-β receptors and SMAD proteins are common in CRC. In a study of 128 metastatic CRC (mCRC) patients, 17% had alterations in the TGF-β pathway [16]. Another study of 579 patients undergoing colorectal liver metastasis (CRLM) resection found TGF-β mutations in 11.2% of cases [19]. High microsatellite instability (MSI-H) CRC often has frequent inactivating mutations in the TGF-βR2 gene, leading to truncated receptors [21].

Mutations in SMAD2 and SMAD4 can disrupt TGF-β signaling, resulting in uncontrolled cell growth and tumor progression. In sporadic CRC, SMAD4, SMAD2, and SMAD3 mutations were found in 8.6%, 3.4%, and 4.3% of cases, respectively, including various types of mutations [22]. SMAD7, an inhibitory protein, is often increased in CRC and is linked to a poor prognosis due to its amplification caused by single nucleotide polymorphisms [23]. Dysregulation of BMPs may also play a role in the development of sporadic CRC [24]. Additionally, the loss of SMAD4 in mCRC, occurring in about 30% of cases, contributes to resistance to chemotherapy [9,25]. Mutations in the TGF-β pathway often occur alongside other signaling pathway variations, suggesting a synergistic effect on CRC metastasis, rather than being the primary drivers [26].

Several non-canonical pathways contribute to epithelial-mesenchymal transition and neoangiogenesis, influencing the tumor microenvironment. These include the mitogen-activated protein kinase (MAPK) pathway, the Rho-associated kinase (ROCK) pathway, and the phosphoinositide 3-kinase (PI3K)/protein kinase B (AKT) pathway [17].

In TGF-β-sensitive intestinal and lung epithelial cells, both TGF-β receptors (TGF-β RI and TGF-β RII) activate RAS, which is crucial for regulating specific cell cycle events and inhibiting the mitogenic response in intestinal epithelial cells. Activation of MAPKs SAPK and Erk via Ras and TGF-β receptors is necessary for the autoinduction of TGF-β RI. Additionally, TGF-β-induced activation of the Ras/MAPK pathway positively influences the Smad1 signaling pathway [27,28] (Figure 1).

## 3. The Role of TGF-β in Epithelial Mesenchymal Transition (EMT)

EMT is crucial during embryogenesis and plays a key role in cancer metastasis. During EMT, cells lose E-cadherin expression and redistribute Zonula Occludens (ZO) proteins, claudins, and occludins, altering cell polarity and reorganizing the cytoskeleton [29]. This transcriptional change activates mesenchymal genes and increases the expression of extracellular proteases, which degrade extracellular matrix proteins and enable invasive behavior [30]. Consequently, cells undergoing EMT acquire a mobile and invasive phenotype [31,32]. Mesenchymal cancer cells are associated with poor prognoses and chemotherapy resistance [33]. A key step in EMT is the loss of E-cadherin, caused by epigenetic modifications like hypermethylation of its gene promoter, leading to its transcriptional silencing. This loss triggers cytoskeletal reorganization and activates signaling pathways, such as TGF-β and Wnt, which further suppress E-cadherin and promote a mesenchymal phenotype. These pathways lead to the upregulation of mesenchymal proteins, such as vimentin, enhancing the invasive capabilities of cancer cells [34,35].

During EMT, increased deposition of extracellular matrix proteins forms focal adhesion complexes that facilitate cell migration [36]. The expression of N-cadherin also increases, making cells more mobile and invasive [37]. These changes result in the loss of apical-basal polarity in epithelial cells, enhancing their migration [38]. TGF-β activates various cellular signals and mechanisms that contribute to tumor progression, promoting EMT and cancer stem cell (CSC) proliferation [39,40,41]. In hepatocarcinoma stem cells, TGF-β increases the expression of the stem cell marker CD133 and the adhesion molecule CD44 via the SMAD-dependent pathway. In glioma stem cells, the same pathway induces self-renewal through the LIF-Janus kinase-STAT pathway [41,42,43]. Both SMAD-dependent and independent pathways influence cell junction complexes [17,44,45]. While similar processes may occur in CRC, further research is needed to confirm these observations.

The role of SMAD4 in tumorigenesis and EMT is complex. SMAD4 can act as a tumor suppressor in some cells and as an EMT inducer in others. TGF-β-induced, SMAD-mediated EMT increases the expression of ZEB proteins and HLH transcription factors, as well as Snail, Slug, and Twist, which repress E-cadherin expression [38,46,47,48].

SMAD-independent pathways also promote EMT by dissolving cell junctions and remodeling the cytoskeleton. This involves both the canonical pathway (inducing Snail and Zeb expression) and the non-canonical pathway (downregulating Par3 and degrading RhoA) [17,49,50]. These changes activate ERK, which forms a complex with SHC and GRB2, enhancing SMAD transcriptional activity and promoting invasion and metastasis [51,52,53].

The PI3K-AKT-mTOR pathway is another non-SMAD pathway involved in TGF-β-mediated EMT. TGF-β activates AKT via PI3K and E3 ligase TRAF6, which also activates JNK and p38 kinases, playing a central role in EMT [54,55]. Additionally, TGF-β upregulates PDGF receptors and ligands, leading to PI3K activation and the initiation of EMT via the SRC/STAT3 pathway [56].

Recent evidence suggests that miRNAs contribute to EMT via TGF-β signaling. TGF-β modulates miRNA expression at both transcriptional and post-transcriptional levels [57,58]. Gregory et al. found that the microRNA-200 family (miR-200a, miR-200b, miR-200c, miR-141, and miR-429) and miR-205 are significantly downregulated in cells undergoing EMT in response to TGF-β. Enforced expression of these miRNAs can prevent EMT [59]. TGF-β influences miR-200 family expression through a TGF-β/ZEB/miR-200 feedback loop mediated by SMAD4. The miR-200 family regulates E-cadherin repressors, so their downregulation leads to the overexpression of ZEB1 and ZEB2, which are transcriptional repressors of E-cadherin. In CRC cells, upregulation of ZEB1 at the tumor’s invasive front is linked to basement membrane loss and EMT [60]. Gregory et al. also demonstrated that the inhibition of miR-200 alone induces a mesenchymal-like phenotype [61,62].

Recent studies show that TGF-β regulates the expression of long non-coding RNAs (lncRNAs), which are increasingly recognized as TGF-β effectors [58]. In CRC, taurine-upregulated lncRNA gene 1 (TUG1) mediates TGF-β-induced EMT, enhancing migration, invasion, and risk of pulmonary metastases via the TWIST1/EMT pathway. However, the regulation of TUG1 by TGF-β is not fully understood [63]. In mCRC, TGF-β signaling via SMAD2 downregulates lncRNA LINC01133, which normally prevents EMT by inhibiting the EMT promoter SRSF6. Conversely, it upregulates lncRNA-ATB, which suppresses E-cadherin expression [64,65].

TGF-β also promotes the generation of myofibroblasts from mesenchymal precursors, aiding in tumor invasion [66]. The TGF-β receptor variant TGF-ΒR1*6A can switch TGF-β1′s role from inhibiting to stimulating growth, significantly increasing cell invasion in mCRC via pathways like Ras/MAPK, JNK, and PI3K/AKT [67]. Zhou et al. found that cells with the TGF-ΒR1*6A allele had increased activity of the p38 and ERK1/2 MAPK pathways, independent of TGF-β itself, potentially due to secondary signaling mediated by the cleaved signal sequence in the cytoplasm [68]. Additionally, the inactivation of TGF-βR2 in intestinal epithelial cells promotes tumor transformation and invasion initiated by APC mutation in a cell-autonomous manner [69].

Cross-talk between Hypoxia Inducible Factor (HIF)-1α and TGF-β has been proposed, especially in renal carcinoma. HIF-1α upregulates TGF-β and activates its SMAD-dependent pathway in early tumor progression. The Von Hippel-Lindau (VHL) protein inhibits ALK5 and HIF-1α/2α under normoxia, while hypoxia increases their expression. ALK5 further enhances HIF-1α/HIF-2α expression under normoxia.

During hypoxia, HIF-1α increases hypoxia-response elements like N-Cadherin through Snail’s transcriptional activity. TGF-β and HIF-1α form a feed-forward loop, enhancing EMT [70]. Preclinical studies with sanguinarine show that inhibiting HIF-1α reduces EMT, Snail translocation, and activation of the Smad and PI3K-AKT pathways [71,72,73] (Figure 2).

## 4. The Role of TGF-β in Angiogenesis

Angiogenesis, a hallmark of cancer, involves the formation of new blood vessels to supply tumors with oxygen and nutrients, facilitating their growth and spread [74,75]. These new vessels also help cancer cells enter the bloodstream and metastasize to distant sites [76].

TGF-β can act as either an angiogenic or angiostatic factor, depending on interactions between tumor cells, normal epithelial cells, and the tumor microenvironment [77]. TGF-β stimulates angiogenesis by accelerating endothelial cell migration and proliferation [78]. It increases VEGF expression in endothelial cells via Sp1-dependent transcriptional activation, promoting proliferation, migration, and survival by binding to VEGFR-1/2 [79,80]. In a mouse model of liver metastases, inhibiting TGF-β-induced ig-h3 protein suppresses CRC cell angiogenesis and inhibits CRC liver metastases progression [81]. TGF-β also promotes pericyte differentiation during blood vessel formation through a SMAD-dependent pathway [82,83]. Additionally, TGF-β activates the Notch1/Twist1 pathway, upregulating platelet-derived growth factor D (PDGF-D), which promotes CRC cell growth, migration, and angiogenesis [84].

TGF-β induces proangiogenic factors such as connective tissue growth factor (CTGF) and insulin-like growth factor-binding protein 7 (IGFBP-7) in epithelial cells and fibroblasts [85]. TGF-β1, through TGFβRII-mediated SMAD2 and p38 pathways, along with thrombospondin 1, induces Runt-Related Transcription Factor-1 (RUNX1) overexpression in mCRC tumor cells, enhancing cell motility for new vessel formation [86].

Conversely, an SMAD4-dependent pathway can act as a tumor suppressor. Downregulation of SMAD4 expression leads to VEGF overexpression, promoting neoangiogenesis and metastasis in mCRC [25]. In pancreatic cancer cells, restoring SMAD4 expression suppresses neoangiogenesis by reducing VEGF and increasing Thrombospondin 1 (TSP1) [87]. (Figure 3).

## 5. The Role of TGF-β in the Tumor Microenvironment

The tumor microenvironment (TME) is crucial for carcinogenesis, progression, immune evasion, and metastasis. TGF-β signaling influences various elements of TME, promoting tumor progression.

Increased TGF-β secretion aids tumor immune evasion. Tauriello et al. [88] showed that higher TGF-β levels in the TME cause T cell exhaustion and inhibit the Th1-effector phenotype, leading to an immunologically cold environment and poor prognosis. Combining TGF-β inhibitors with anti-PD1/PDL1 treatment increases lymphocyte infiltration and T-bet expression and reduces T-cell exhaustion, enhancing the immune response and improving susceptibility to anti-PD1/PDL1 drugs.

Increased TGF-β signaling also promotes the conversion of fibroblasts into cancer-associated fibroblasts (CAFs), accelerating disease progression and metastasis [89]. Peng et al. [90] demonstrated that integrin αvβ6 expression in CRC cells induces fibroblast conversion, marked by increased α-SMA and FAP expression.

TGF-β plays a pivotal role in activating CAFs, which promote CRC metastasis through several mechanisms. Activated CAFs secrete SDF-1, activating the SDF-1/CXCR4 axis, thereby enhancing CRC cell metastasis [91]. Additionally, IL11, secreted by TGF-β-activated CAFs, triggers GP130/STAT3 signaling in CRC cells, providing a survival advantage for metastatic cells [91]. An important study from Spain in 2015 [92] demonstrated that increased TGF-β expression enhances adhesion between CAFs and CRC cells, leading to increased proliferation, invasion, and liver metastases in vivo. Inhibiting TGF-β with P17 blocks CAF-CRC cell adhesion and reduces liver metastasis in experimental models [92]. Furthermore, CAFs can self-stimulate through a positive feedback loop. Zhang et al. [93] found that miR-17-5p produced by CAFs increases TGF-β production and secretion by CRC cells. In turn, TGF-β stimulates CAFs to produce more miR-17-5p through the RUNX3/MYC/TGF-β1 pathway, promoting a metastatic phenotype.

TGF-β undergoes regulatory mechanisms to control its expression; recent findings indicate that Thrombospondin Type 1 Domain Containing 4 (THSD4) can hinder TGF-β binding to its receptor by interacting with microfibrils [94,95]. Conversely, Zinc Finger Protein 37A (ZNF37A), often upregulated in undifferentiated CRCs, inhibits THSD4 transcription by binding to its promoter, thereby promoting TGF-β signaling and facilitating stromal fibroblast conversion into CAFs, which enhances tumor metastasis [89].

Tumor-associated macrophages (TAMs) play a crucial role in the tumor microenvironment (TME), as the predominant infiltrating immune cells. TAMs are known promoters of cancer metastasis, although the specific mechanisms involved are not fully understood [96,97,98]. TGF-β produced by TAMs has been shown to activate HIF1α through glycolysis, which subsequently activates the TRIB3/β-catenin/Wnt signaling pathway, thereby enhancing tumor progression. Inhibition of HIF1α signaling by GN44028 has demonstrated potential anti-cancer effects when combined with chemotherapy, reducing tumor progression [99].

GDF15, a member of the TGF-β superfamily secreted by macrophages, promotes invasion and metastasis by enhancing ERK 1/2 phosphorylation of c-Fos [100]. Hypersecretion of TGF-β by TAMs in response to oxaliplatin-based chemotherapy induces PDL1 upregulation, fostering an immunosuppressive tumor microenvironment. Targeting TGF-β or its upstream regulators to inhibit PDL1 expression could potentially enhance CRC sensitivity to chemotherapy [101].

CRC cells stimulate M2-type macrophage polarization, characterized by an inflammatory and pro-tumorigenic phenotype, through the secretion of CTHRC1 and activation of the TGF-β signaling pathway. CTHRC1 interacts directly with TGFβR-2 and -3, stabilizing the ligand-receptor complex [102]. M2-type TAMs, in turn, promote Tregs generation via the TGF-β/SMAD signaling pathway [103], and induce EMT through the SMAD2,3-4/Snail/E-cadherin signaling pathway [99], thereby enhancing metastasis.

TME encompasses various cell components, including tumor-associated neutrophils, myeloid cells, monocytes, and T cells, all positively correlated with CRC progression and metastasis through distinct mechanisms. Neutrophil infiltration, induced by TGF-β and NOTCH1 activation, enhances liver metastasis [104]. Additionally, TGF-β upregulation correlates with the increased expression of Th17 and Treg-related genes, promoting cytokine release that contributes to CRC progression [105]. For a summary, see Figure 4.

## 6. Additional Roles of TGF-β in mCRC Development

Recent evidence indicates that radiation increases TGF-β production by CRC cells, leading to elevated expression of podocalyxin-like protein (PODXL) and enhanced deposition of the extracellular matrix. This effect strengthens the migratory and invasive properties of cancer cells. Inhibition of the TGF-β pathway by Galunisertib in CRC cells exposed to radiation suppresses PODXL activation, thereby inhibiting cell invasion and migration [106].

Downregulation of the TGF-β pathway contributes to the formation of tumor spheres with inverted polarity (TSIPs) through the ROCK pathway, particularly involving SMAD2. Unlike normal epithelial tissue, TSIPs exhibit inverted apical-basolateral polarity, with the apical pole oriented outward. Recent studies have linked TSIPs to the process of CRC metastasis to the peritoneum [107].

In MSI-H CRC, mutations in the TGF-βR2 gene are frequent, occurring in approximately 74% of cases. These mutations are associated with increased vascular invasion, which contributes to tumor progression due to TGF-β-mediated inhibition of epithelial cell growth. Lack of responsiveness to TGF-β signaling enhances cell growth and invasion [21]. Importantly, there are no statistically significant differences in overall survival between MSI-H patients with TGF-βR2 mutations and those without [108].

Moreover, TGF-βR2 mutations are often found alongside alterations in other signaling pathways in mCRC. The accumulation of these mutations synergistically enhances the metastatic process of CRC. For instance, the Kirsten rat sarcoma virus (KRAS) G12D mutation, commonly co-occurring with TGF-βR2 mutations in mCRC, induces EMT-like cell morphology and increases the incidence of liver metastasis in animal models.

In mouse models, the presence of only the KRAS G12D mutation does not lead to an increase in liver metastasis. This underscores the critical role of the TGF-β pathway as a mediator of invasiveness, specifically in KRAS mutant mCRCs [109].

Mutations in the APC gene, commonly found in patients with familial adenomatous polyposis (FAP), are associated with deregulation of the Wnt signaling pathway. Studies indicate that dysregulation of both TGF-β and Wnt signaling pathways synergistically promotes colorectal tumorigenesis [110].

Carcinoembryonic antigen (CEA) serves as a tumor marker in mCRC, and its periodic measurement is a routine part of clinical assessment during both the follow-up and active treatment phases. Elevated CEA levels influence the abnormal activation of the TGF-β pathway through the MAPK-NFκB pathway and HNRMP/CEAR, ultimately converging on NF-κB [111] (Figure 5).

## 7. TGF-β as a Therapeutic Target

TGF-β plays a critical role in CRC progression by promoting EMT and angiogenesis and by creating an immunosuppressive microenvironment. Targeting TGF-β signaling is a promising therapeutic approach under investigation.

Several monoclonal antibodies have been explored in this regard, albeit with mixed outcomes. SAR4349459 initially showed promise but was discontinued due to toxicity and a low objective response rate [112,113]. Another monoclonal antibody, PF-03446962, which blocks TGF-βRI, was studied in various tumors but demonstrated unacceptable toxicity and limited clinical activity in a phase Ib trial (REGAL-1) when combined with Regorafenib in CRC [114].

Bintrafusp alfa, combining PDL1 and TGF-βRII inhibition, demonstrated clinical activity but yielded mixed results in CRC patients, particularly effective in CMS4 subtype cases [115]. However, trials in oligometastatic CRC and MSI-H cancers did not show significant anti-tumor effects [116,117].

NIS793, a TGF-β-targeting monoclonal antibody, combined with spartalizumab, an anti-PD1 antibody, showed promising results in a phase Ib trial with advanced solid tumors. Among 60 patients, partial responses were seen in renal cell carcinoma and MSS-CRC cohorts, with no dose-limiting toxicity reported. Evidence included increased TGF-β/NIS793 complexes and reduced active TGF-β levels in peripheral blood. Tumor biopsies showed decreased TGF-β target gene expression and enhanced immune signatures [118].

Dalutrafusp alpha (GS-1423), targeting CD73-adenosine production and TGF-β signaling, was well tolerated in a phase I study of CRC patients. Grade 3 or 4 adverse events occurred in 42.9% of patients, with two cases of Grade 5 events (pulmonary embolism and progressive disease) deemed unrelated to dalutrafusp alpha. Responses varied among the four CRC patients, with mixed responses observed [119].

Most clinical trials targeting TGF-β in humans have fallen short of expectations. This discrepancy with preclinical data could be attributed to several factors. Timing of administration differs significantly between preclinical and clinical trials: in preclinical studies, anti-TGF-β agents are initiated early, whereas in clinical trials, they are often used in heavily pre-treated and advanced patients, limiting efficacy. Moreover, animal models may not fully reflect human conditions, especially concerning components like cancer-associated fibroblasts (CAFs) and their interaction with TGF-β in the extracellular matrix. The TGF-β pathway is intricately regulated by feedback mechanisms; therefore, stopping TGF-β inhibition may lead to increased receptor responsiveness, resembling a flare-up scenario. Other strategies are still in the preclinical stages of investigation.

Kinase inhibitors have shown promising initial results in both in vitro and in vivo studies. LY2109761, a TGF-β receptor inhibitor, effectively blocked liver metastasis in mouse models [120]. Combining Galunisertib (LY2157299) with adoptive NK cells also resulted in the eradication of liver metastases [121]. Furthermore, Galunisertib combined with an AXL inhibitor, Bemcentinib, reduced colony formation and migration in mesenchymal-type human CRC cell lines [122].

lncRNAs are RNA transcripts over 200 nucleotides long that do not translate into proteins [123]. While lncRNA therapies are not yet standard, they show promise as drug targets due to their tissue specificity, potentially reducing off-target effects [124]. For instance, MIR503HG and HOXC-AS3 have been identified as upstream inhibitors of TGF-β2; their overexpression inhibits the migration and invasion of CRC cells [125]. Conversely, silencing lncRNAs like EZR-AS1 blocks TGF-β signaling, stimulates apoptosis, and reduces migration [126]. Similarly, targeting LINC00941 theoretically could inhibit migration and invasion [127].

Sitagliptin, a DPP-4 inhibitor used in type 2 diabetes, was investigated by Varela-Calviño et al. in 2021 for its effects on CD26+ (DPP4) CRC cell lines. They found that Sitagliptin not only antagonizes DPP-4 directly but also interferes with TGF-β1 effects on EMT and cell cycle, thus limiting metastasis [128] (Table 1).

However, Principe et al. (2016) highlighted the potential drawbacks of TGF-β inhibition. Loss of TGF-β signaling could lead to fatal inflammatory diseases in APC mice and might accelerate carcinogenesis [129]. A clinical trial with LY3022859, an anti-TGF-βR2 monoclonal antibody, was halted prematurely due to a high toxicity profile, including cytokine release syndromes, despite prophylactic use of antihistamines and corticosteroids [130].

Principe et al. demonstrated in mouse models that genetically blocking TGF-β signaling led to increased tumor-associated inflammation, tumor burden, enhanced tumor development, and increased mortality [129]. This dual role of TGF-β in carcinogenesis is evident in many tumors, where there is a gradual shift from its growth-suppressive effects (reduced proliferation, increased apoptosis) to its pro-migratory signaling, promoting invasive and metastatic behavior. Loss of SMAD4 in CRC correlates with poorer survival, yet in advanced cancer stages, TGF-β is often overexpressed and serves as a negative prognostic factor [131,132,133].

## 8. TGF-β and Mechanism of Resistance to Chemotherapy and Biological Agents

An association between TGF-β expression and resistance to chemotherapy has also been reported [134]. TGF-β expression has been linked to chemotherapy resistance in CRC. High levels of HIF-1α/TGF-β2 correlate with tumor relapse post-chemotherapy, indicating these factors as potential targets for overcoming chemo-resistance [135].

TGF-β contributes to oxaliplatin resistance in mCRC by suppressing macroautophagy via the TGF-β/SMAD4 axis and by increasing EMT, which reduces DNA damage and apoptosis induced by oxaliplatin [136]. Additionally, abnormal TGF-β receptor expression can protect CRC cells from 5-FU’s cytotoxic effects. SMAD4-mediated transcription increases resistance to 5-FU, whereas inhibiting TGF-β signaling restores chemosensitivity. Chemoresistant cells often lack functional TP53, affecting TGF-β’s role in cytostasis, ECM functioning, and metastasis, thus influencing drug resistance [137,138].

In BRAF mutant cells, TGF-β signaling is often up-regulated after BRAF inhibitor (BRAFi) treatment, leading to resistance [139,140]. Sun et al. found that suppressing SRY-box transcription factor 10 (SOX10) in BRAFi-treated melanoma cells activates TGF-β signaling, upregulating EGFR and PDGFRb, and confers resistance to MAPK inhibitors [141]. BRAFi-treated melanoma cells show increased EGFR, PDGFRB, and miR-125a expression due to TGF-β signaling, reducing pro-apoptotic pathway activity and fostering BRAFi resistance [141,142]. Conversely, BRAF mutant cells may depend on TGF-β signaling, enhancing TGF-β inhibitors’ efficacy [143]. Additionally, TGF-β overexpression contributes to cetuximab resistance in CRC cells by inducing the EGFR–MET interaction [144].

In conclusion, targeting TGF-β alone has limitations, but combining TGF-β inhibition with other anti-cancer agents can reduce chemoresistance (Figure 6). In a phase 2 trial, Galunisertib with neoadjuvant chemoradiotherapy in locally advanced rectal cancer showed promising results. Out of 38 patients, 25 (71%) completed the treatment and underwent surgery, with 5 (20%) achieving pathological complete responses (pCR). Ten patients (29%) were initially ineligible for surgery, but 3 (30%) chose to undergo it later, with 2 (67%) achieving pCR. Of the 7 patients opting for non-operative management, 5 (71%) had clinical complete responses (cCR) at 1 year after their last modified FOLFOX6 infusion. Overall, 12 (32% [one-sided 95% CI ≥ 19%]) patients had CR. Only 2 (5%) patients experienced grade 4 adverse events. These results suggest that TGF-β inhibition with Galunisertib enhances sensitivity to chemoradiation in patients with locally advanced rectal cancer, improving the complete response rate to 32% [145].

## 9. Conclusions

TGF-β pathway is a key element in mCRC, involved in promoting EMT, favoring angiogenesis and interaction with the tumor microenvironment. Thus, it represents a potential target of intervention on both CRC cells and their microenvironment. Unfortunately, no TGF-β inhibitor is available, at least to our knowledge, in clinical practice. Clinical trials in humans have often failed expectations, probably due to differences in timing and the difficulty in switching from an animal to human models. Better patient selection, perhaps through identification of biomarkers, and further understanding of TGF-β pathway and regulation mechanisms may help in the development of effective strategies. In this view, the patients enrolled in clinical trials rarely have been characterized for serum TGF-β levels of for mutations in the TGF-β-signaling. Moreover, due to the low number of patients enrolled in the study, with TGF-β inhibitors, no information about additional molecular markers of response are available. Another consideration has to be about the possible use of TGF-β inhibitors in combination with chemo- and/or radiotherapy that have been demonstrated to be potent inducers of EMT remodeling through TGF-β signaling modulation, as reported above. Therefore, the inhibition of the TGF-β pathway could be useful in order to potentiate their efficacy. Future studies should be performed using TGF-β inhibitors in combined strategies with chemo- and radiotherapy approaches.

Moreover, TGF-β dysregulation occurs much later in CRC, at locally advanced or metastatic stages. A result of this dysregulation is increased TGF-β secretion from other stromal cells rather than the tumor cells themselves. Because PD-L1 can be upregulated on both tumor cells and infiltrating immune cells, this creates a target-rich environment for TGF-β inhibitors and thus more opportunity to provide localized TGF-β inhibition in the tumor. This encourages the combined use of TGF-β inhibitors with PD1/PDL1 inhibitors in defined clinical settings after molecular characterization of the tumors [146]. The toxicity of TGF-β must also be kept in mind when evaluating potential therapeutic benefits. Therefore, further research is definitely needed to finally target TGF-β in clinical practice in order to identify better drug combinations and improve patient selection.

## Figures and Tables

**Figure 1 ijms-25-07400-f001:**
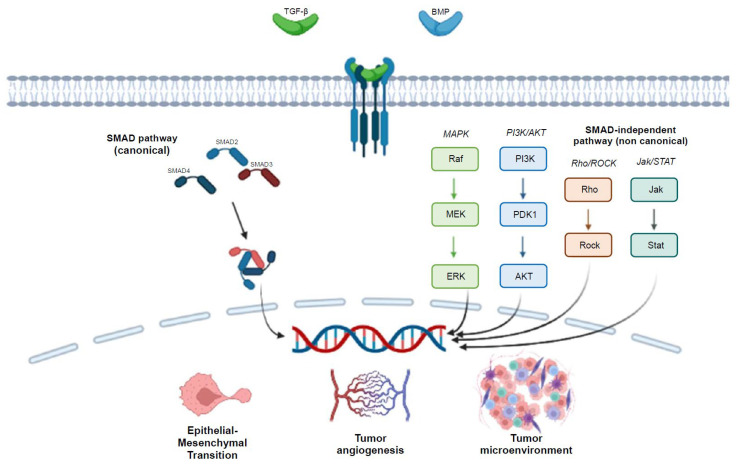
A schematic representation of the TGF-β signaling pathway illustrates its key components and interactions in both canonical and non-canonical pathways.

**Figure 2 ijms-25-07400-f002:**
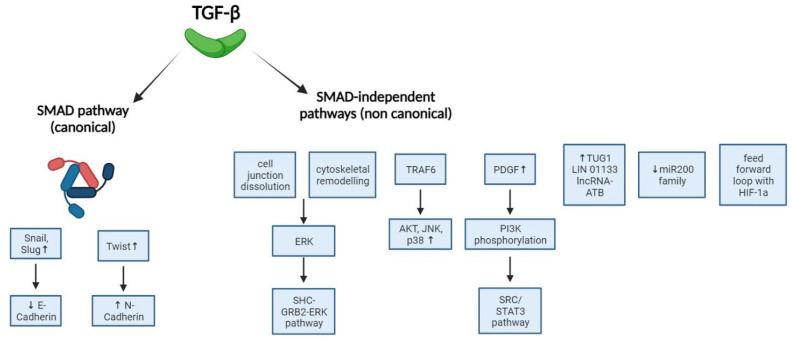
The role of TGF-β in EMT: SMAD-dependent and SMAD-independent pathways.

**Figure 3 ijms-25-07400-f003:**
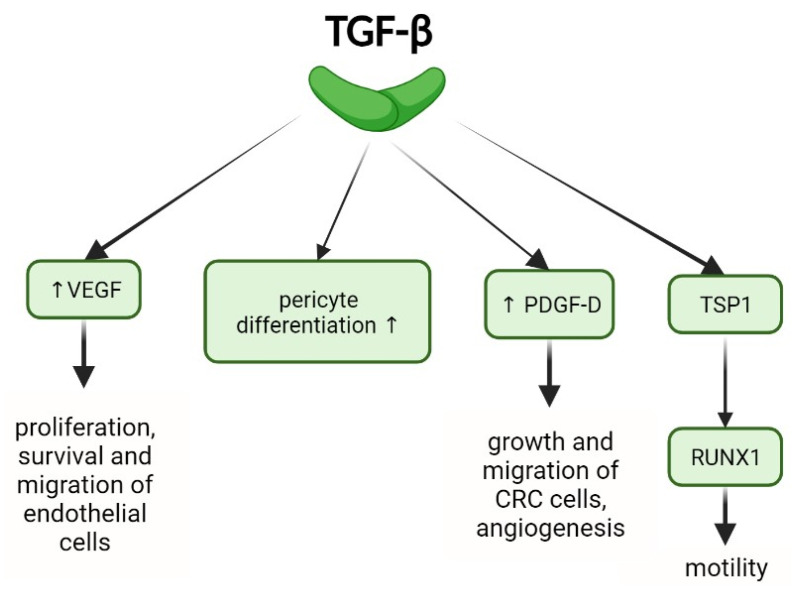
The role of TGF-β in angiogenesis: pericyte differentiation, and regulation of key angiogenic factors including VEGF, PDGF-D, and TSP-1.

**Figure 4 ijms-25-07400-f004:**
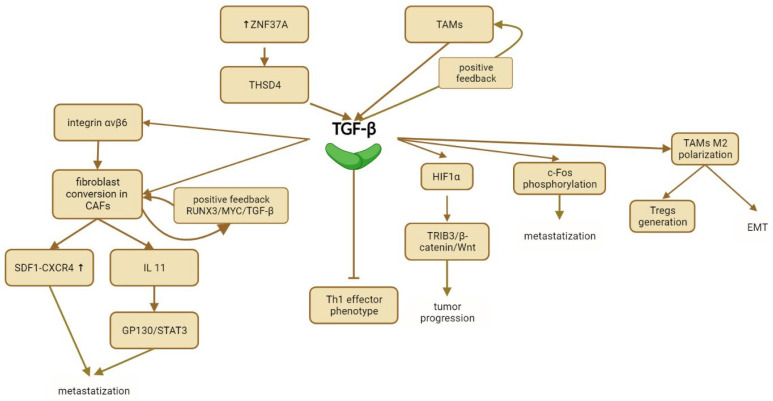
Interactions of TGF-β with the tumor microenvironment: insights into TGF-β signaling, EMT, metastasis, and tumor progression.

**Figure 5 ijms-25-07400-f005:**
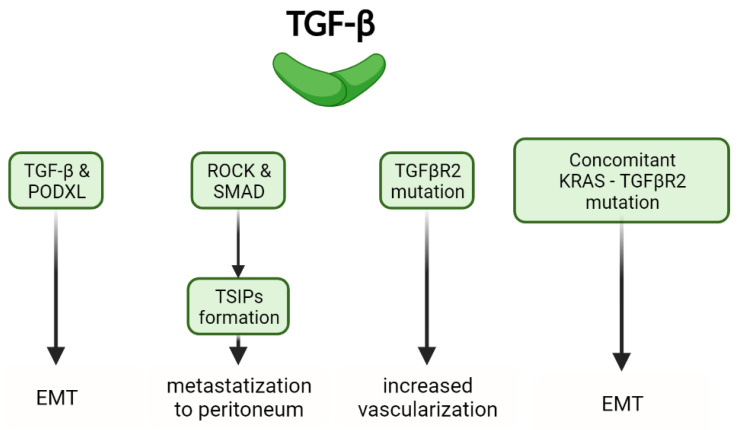
Additional roles of TGF-β: EMT regulation by TGF-β and PODXL, peritoneal metastasis through TSIP formation via ROCK and Smad, TGFβR2 mutations in vascularization enhancement, and concomitant KRAS-TGFβR2 mutation in EMT.

**Figure 6 ijms-25-07400-f006:**
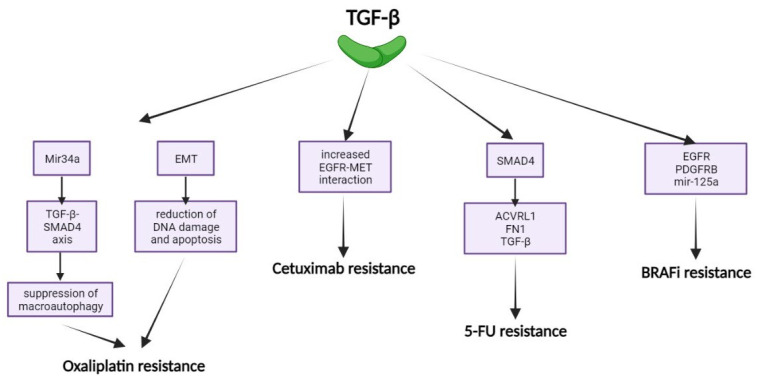
TGF-β mediated mechanisms of resistance: Oxaliplatin resistance, Cetuximab resistance, 5-FU resistance, BRAFi resistance.

**Table 1 ijms-25-07400-t001:** Trials with TGF-Β inhibitors (AEs: adverse events; MSI-H, microsatellite instability-high; EMT, epithelial-mesenchymal transition).

Drug	Trial	Setting	Results
SAR4349459	NCT03192345 [112,113] (±cemiplimab)	Phase 1/1b	Around 50% ≥ G3 AEs; terminated due to low objective response
PF-03446962	REGAL-1 [114] (+Regorafenib)	Phase 1b	No clinical activity, unacceptable toxicities
Bintrafusp alfa	NCT02517398 [115]	Phase I	Only 1 CMS4 patient presented long response,
	NCT03436563 [116]	Phase Ib/II	Treatment determined increased progression
	NCT03436563 [117]	Phase Ib/II	No significant activity in MSI-H mCRC
LY2109761	[120]	Preclinical	Reduced liver metastatization
Galunisertib	[121]	Preclinical	Eradication of liver metastasis
	[122] (+Bemcentinib)	Preclinical	Reduced colony formation and migration
Sitagliptin	[128]	Preclinical	Limiting EMT and metastatization

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
