# Peer review of "TGF-β Modulated Pathways in Colorectal Cancer: New Potential Therapeutic Opportunities"

_ijms, 2024, doi:10.3390/ijms25137400_

Round 1

Reviewer 1 Report (Previous Reviewer 1)

Comments and Suggestions for Authors

The manuscript has not been significantly improved compared to previous versions. I confirm my former decision

Comments on the Quality of English Language

English should be accurately revised. I detected several mistakes and repetitions. 

Author Response

Reviewer 1

Comments from Reviewer 1

There are some parts that still need to be improved:

-Lines 102-104: it should be specified the functional effect(s) of the cited mutations.

Answer: We thank the reviewer for this warning, and we have included further clarifications in the text of the revised version of the manuscript as follows:

Mutations and alterations in SMAD proteins, such as SMAD2 and SMAD4, are particularly relevant in the advanced stage CRC. Mutations in these genes can compromise the TGF- β signaling pathway, leading to dysregulation of cell growth, loss of tumor suppression, and tumor progression in advanced CRC. Additionally, mutations in the coding regions for SMAD proteins can influence the formation of the signal transduction complex, further compromising intracellular signal transduction and contributing to tumor growth and metastasis in advanced CRC”.

- Lines 104-106: these two sentences could be better linked

Answer: We have better connected the two sentences as follows:

“The prevalence of mutations in SMAD4, SMAD2, and SMAD3 was found to be 8.6%, 3.4%, and 4.3%, respectively, in sporadic CRC, encompassing somatic truncations, frameshift, missense, and splice site mutations [22]. Additionally, SMAD7, an inhibitory protein that antagonizes TGF-β signaling, shows a significant increase in CRC. Single nucleotide polymorphisms within the SMAD7 gene lead to its amplification and are associated with a poor prognosis [23] Dysregulation of BMPs may also play a role in the development of sporadic CRC [24]. The loss of SMAD4 in mCRC occurs in approximately 30% of cases [9]. Furthermore, SMAD4 loss contributes to resistance to 5-fluorouracil-mediated apoptosis in at least two cell lines [25]. However, mutations in the TGF-β pathway often coincide with variations in other signaling pathways. This suggests that the accumulation of these mutations in mCRC synergistically affects CRC metastases but may not be a primary driver of oncogene addiction [26].”

- Lines 117-122: this section needs more details. Which cell cycle events are specifically regulated by RAS/TGF-b? What do the authors mean with "autoinduction" of TGF-b RI?

Answer: We added more details as suggested.

Regarding the meaning of autoinduction, in the context of TGF-β RI, autoinduction implies that the signaling cascade initiated by TGF-β leads to the upregulation or increased expression of TGF-β RI receptors themselves.

- Lines 139-141: how does E-cadherin loss induce the mentioned phenomena? What are the molecular mechanisms?

Answer: We thank the reviewer for this suggestion and have added the molecular mechanisms in the revised form of the manuscript as follows:

“The loss of E-cadherin, mediated by epigenetic modifications and alterations in gene transcription, represents a critical step in the process of EMT. This loss initiates a cascade of molecular events within epithelial cells, including cytoskeletal reorganization and activation of signaling pathways. Epigenetic modulation, such as hypermethylation of the E-cadherin gene promoter, leads to transcriptional silencing of E-cadherin expression. Concurrently, transcription factors involved in EMT are upregulated, further suppressing E-cadherin expression and promoting the acquisition of a mesenchymal phenotype. Additionally, the loss of E-cadherin activates signaling events that drive the transition towards a mesenchymal identity. Signaling pathways such as TGF-β and Wnt are often activated in response to E-cadherin loss, further promoting EMT. These signaling pathways converge on downstream effectors, leading to the upregulation of mesenchymal cytoskeletal proteins, including vimentin.”

- Lines 147-150: the mentioned mechanisms do not occur in CRC. There are some hints that similar processes could occur also in CRC? Otherwise, the whole sentence is misleading, and the authors should clarly specify the systems in which observations have been made

Answer: We thank the reviewer for this observation. In the revised version of the manuscript we have clarified and rewritten entire sentences as follows:

The effects of TGF-β activation on tumor progression have been primarily investigated in contexts such as hepatocarcinoma and glioma, where specific TGF-β activation mechanisms have been observed. In hepatocarcinoma stem cells, TGF-β upregulates the stem cell marker CD133 and the adhesion molecule CD44 via the SMAD-dependent pathway, while in glioma stem cells, the same pathway induces self-renewal through the LIF-Janus kinase-STAT pathway [41–43]. Furthermore, a combination of both SMAD-dependent and independent effects influences cell junction complexes [17,44,45]. Although there are indications that similar processes could potentially occur in CRC, further research is needed to confirm these observations."

- Lines 211-220: I recommend the authors to use a homogeneous nomenclature for molecules. Here, ALK5 could not be immediately associated to TGF-b RI by the reader. Moreover, the reference refers to renal carcinoma and not CRC. The authors should clearly mention that these evidences have not been observed in CRC ad thus the possibility that this mechanism also occurs in CRC is just speculative

Answer: Thank you for your suggestion. We acknowledge the importance of using a consistent nomenclature for molecules to improve clarity and understanding. We will ensure uniformity in our terminology throughout the manuscript. Regarding the reference to ALK5, we will explicitly state that ALK5 refers to the TGF-β type I receptor to avoid ambiguity. Additionally, we clarify that the cited reference pertains to renal carcinoma rather than CRC as follows:

“A cross-talk between Hypoxia Inducible Factor (HIF)-1α and TGF-β has been proposed, particularly observed in renal carcinoma. HIF-1α is suggested to contribute to the upregulation of TGF-β and activation of its SMAD-dependent pathway, especially in the earlier phases of tumor progression, as observed in renal carcinoma. Additionally, it has been observed that Von Hippel-Lindau (VHL) protein inhibits the expression of ALK5, which is mediated by TGF-β, and the expression of HIF-1α/2α under normoxic conditions. Conversely, hypoxia increases the expression of HIF-1α/HIF-2α and ALK5. Notably, the latter further enhances the expression of HIF-1α/HIF-2α under normoxic conditions. During hypoxia, HIF-1α increases the expression of hypoxia-response elements, such as N-Cadherin, through the transcriptional activity of Snail. Additionally, TGF-β interacts with HIF-1α, creating a feed-forward loop between the two factors, which further enhances the development of EMT [70]. Preclinical studies with sanguinarine in various tumor types have demonstrated that inhibition of HIF-1α reduces EMT, Snail translocation, and the activation of the Smad and PI3K-AKT pathways [71–73].”

We believe that these findings support the potential targeting of this pathway in colon cancer and strengthen both the description of the mechanisms and the hypothesis regarding therapeutic strategies in our manuscript.

-Lines 343-345: this sentence is not clear

Answer: We have rewritten the sentences as suggested as follows:

“In MSI-H CRC, the gene encoding TGF-βR2 exhibits a notably high frequency of inactivating mutations, estimated to be around 74%. Tumors harboring mutations in TGF-βR2 demonstrate a greater degree of vascular invasion, thereby contributing to tumor progression.”

- Lines 470-474: this section is unclear and needs to be rewritten

Answer: We have rewritten the sentences as suggested as follows:

“In melanoma cells treated with BRAFi, TGF-β signaling has been observed to increase the expression of EGFR, PDGFRB, and miR-125a, while simultaneously reducing activity in the pro-apoptotic pathway. This upregulation of TGF-β signaling favors resistance to BRAFi treatment [141,142]. Conversely, BRAF mutated cells have been noted to become reliant on TGF-β signaling, potentially enhancing the efficacy of TGF-β inhibitors [143].”

Reviewer 2 Report (Previous Reviewer 3)

Comments and Suggestions for Authors

Dear Authors, 

I would like to sincerely congratulate you for the submitted manuscript. The review of TGFb in CRC development is significant, of current interest, and relevant to both prevention and therapy. 

While I find the review well written, I would kindly ask the authors to assess if they refer to a “TGFb pathway” or several “TGFb-modulated pathways”. This is of high relevance for the title and story of the article. The growth factor does indeed reverse many CRC-altered gene expression in several pathways (all involved in colorectal carcinogenesis), downstream of the ligand binding the growth factor.

Therefore, I would like to suggest a title change based on the above comment. 

References are pertinent and up to date. 

Schemes are extremely clear, facilitating the understanding of the topic. 

All in all, I would like to suppot the manuscript for publication.

Best regards, 

Your Reviewer

Comments on the Quality of English Language

English language should be checked for minor errors. 

Author Response

Reviewer 2

- I would like to sincerely congratulate you for the submitted manuscript. The review of TGFb in CRC development is significant, of current interest, and relevant to both prevention and therapy.

Answer: We thank the reviewer for the appreciation to our work.

While I find the review well written, I would kindly ask the authors to assess if they refer to a “TGFb pathway” or several “TGFb-modulated pathways”. This is of high relevance for the title and story of the article. The growth factor does indeed reverse many CRC-altered gene expression in several pathways (all involved in colorectal carcinogenesis), downstream of the ligand binding the growth factor.

Therefore, I would like to suggest a title change based on the above comment.

-Answer: We thank the reviewer for this good suggestion and have changed the title as follows:

TGF-β modulated pathways in Colorectal Cancer: new Potential Therapeutic Opportunities

- References are pertinent and up to date.
Schemes are extremely clear, facilitating the understanding of the topic.
All in all, I would like to support the manuscript for publication.

Answer: We thank the reviewer for all the positive comments.

This manuscript is a resubmission of an earlier submission. The following is a list of the peer review reports and author responses from that submission.

Round 1

Reviewer 1 Report

Comments and Suggestions for Authors

The current version of the manuscript does not show any significant improvement compared to the previous versions. Althought the authors included some new findings, the manuscript still does not show any significant differences compared to other reviews in the field. Moreover, the text in itself suffers of important flaws. It still looks like as a list of evidences and mechanisms, withouth a comprehensive view; many pathways, which should be the focus of the review, are just mentioned but not described in detail; it is unclear what is the opinion of the authors about TGFb targeting in clinical setting. 

Comments on the Quality of English Language

Quality of English should be improved a little bit.

Author Response

We revised the manuscript according to the previous and more recent comments, rephrasing sections in order to increase internal coherence and highlight the main aspects of our work. While no all pathways received the same focus, we tried to give a comprehensive reviews of main significant elements of TGFb signalling acting as a summary of recent literature. We analysed the different roles of TGFb pathway in mCRC development and its potential therapeutic applications.

Reviewer 2 Report

Comments and Suggestions for Authors

Journal of International Journal of Molecular Sciences

Review Article;

The article entitled “TGF-β Pathway Role in Colorectal Cancer and Potential Therapeutic Opportunities’’. The author review TGF-β signaling as critical role  in cell growth, differentiation, apoptosis, epithelial-mesenchymal transition (EMT), regulation of extracellular matrix, angiogenesis and immune responses. As TGF-β pathway and its alterations are a key element in Colorectal Cancer. While TGF-β acts as a tumor suppressive factor in the early stage of tumorigenesis, after cancer cells become refractory to its inhibiting signals, TGF-β contributes to EMT and proliferation; mutations in TGF-β and SMAD proteins are indeed quite frequent in mCRC. Furthermore, TGF-β can also promote neoangiogenesis via VEGF overexpression, pericyte differentiation and other mechanisms. At last, TGF-β maintains an influence over several elements of the tumor microenvironment, such as T cells, fibroblasts, macrophages, favoring immunosuppression and metastatization. Given its strategic role in multiple directions, different strategies are under scrutiny to target TGF-β in mCRC patients, in order to provide new therapeutic options in the near future in Colorectal Cancer.

I carefully read the manuscript and I accept the manuscript for possible publication in IJMS. There are some common mistakes, references and English language problems in the article which should be corrected by the authors. The author needs to critically revise the manuscript and include some more data. After correcting the mistakes, the article could be considered for publication in the prestigious International Journal of Molecular Sciences Journal.

Comments for Authors

Ø  Write keywords in alphabetical order.

Ø  “Abstract section” The author needs to revise the abstract and explain more about the idea of the study.

Ø  The author needs to include the role of TGF-β in metastasis and in hypoxia formation. Which is the key problem during the treatment of cancer.

Ø  “However, a work published by Principe et al in 2016 also demonstrated some 390 unwanted effects of TGF-β inhibition: loss of TGF-β signalling could lead to fatal 391 inflammatory disease in APC mice and might even accelerate carcinogenesis. [125]. 392 Indeed, a trial in patients with advanced solid tumors with LY3022859, a monoclonal 393 antibody anti TGF-β receptor type II, was prematurely interrupted because of high 394 toxicity profile with cytokine release syndromes, prophylaxis with antihistamines and 395 corticosteroids notwithstanding.[126]” could the author need to explain more about the effect of TGF-β inhibition on normal development and on tumor conditions?

Ø  There are some grammatical mistakes, the author needs to revise the manuscript.

Ø The author discusses Therapeutic Opportunities but needs to explain the agents used for TGF-β. The author needs to give a table to explain the agents for the Therapeutic action against TGF-β.

Cite the following references;

v  https://doi.org/10.1038/s41419-019-2173-1

v  DOI: 10.2174/1871520622666220831124321

v  https://doi.org/10.1016/j.phymed.2021.153500

Author Response

Ø  Write keywords in alphabetical order. Corrected

Ø  “Abstract section” The author needs to revise the abstract and explain more about the idea of the study. Abstract has been revised.

Ø  The author needs to include the role of TGF-β in metastasis and in hypoxia formation. Which is the key problem during the treatment of cancer. Added paragraph about TGFb interaction with HIF1a

Ø  “However, a work published by Principe et al in 2016 also demonstrated some 390 unwanted effects of TGF-β inhibition: loss of TGF-β signalling could lead to fatal 391 inflammatory disease in APC mice and might even accelerate carcinogenesis. [125]. 392 Indeed, a trial in patients with advanced solid tumors with LY3022859, a monoclonal 393 antibody anti TGF-β receptor type II, was prematurely interrupted because of high 394 toxicity profile with cytokine release syndromes, prophylaxis with antihistamines and 395 corticosteroids notwithstanding.[126]” could the author need to explain more about the effect of TGF-β inhibition on normal development and on tumor conditions? Details have been added

Ø  There are some grammatical mistakes, the author needs to revise the manuscript. We revised the manuscript

Ø The author discusses Therapeutic Opportunities but needs to explain the agents used for TGF-β. The author needs to give a table to explain the agents for the Therapeutic action against TGF-β.Added  Table 1.

Cite the following references: https://doi.org/10.1038/s41419-019-2173-1 /  DOI: 10.2174/1871520622666220831124321 /  https://doi.org/10.1016/j.phymed.2021.153500. Citations added

Reviewer 3 Report

Comments and Suggestions for Authors

Dear Authors, 

I would like to acknowledge the hard work that was put into your manuscript. Nevertheless, there are some points to be addressed before it can be ready for publication. 

Abstract.

1. mCRC is an abbreviation not explained before usage in the abstract. 

Introduction. 

1. Up to row 56, the Introduction is rather unfocused for the topic of the article. 

2. In the last paragraph it is inferred that the authors did tests in order to analyze the TGF-ß physiological role. 

Section 2. 

1. This section needs rephrasing in a less convoluted manner. A red string needs to be followed in between paragraphs. 

2. Figure 1 needs to be of higher quality. Please make sure you have the consent of the original authors for any figure you are borrowing from previously published material. 

Section 3. 

1. Please use full name before using abreviations (ZO – line 130)

2. Please make sure you have the consent of the original authors for any figure you are borrowing from previously published material. 

3. This section needs rephrasing in a less convoluted manner. A red string needs to be followed in between paragraphs.

Section 5. 

1. TME is an abbreviation not explained before its usage

2. Please make sure you have the consent of the original authors for any figure you are borrowing from previously published material. 

3. Figure 4 has a missing arrowhead in the positive feedback loop involving the RUNX3/MYC/TGF-ß pathway. 

Sections 6,7&8

1. Please try to follow a line of argumentation with the information provided. 

2. Please summarize each section into a figure, as provided for previous sections. This would rapidly increase readers’ understanding. 

OVERALL REMARKS: 

A proofreading of the manuscript is necessary. There are spelling, punctuation, and formatting mistakes throughout the entire manuscript. Please organize the information provided in the review so that the reader can get the most important parts in the most optimal manner. 

A review is meant to be an unbiased logical summary of a topic. Please keep that in mind when rephrasing your manuscript. 

Best regards, 

Your Reviewer

Comments on the Quality of English Language

A proofreading of the manuscript is necessary. There are spelling, punctuation, and formatting mistakes throughout the entire manuscript. 

Author Response

Abstract.

  1. mCRC is an abbreviation not explained before usage in the abstract. Corrected

Introduction. 

  1. 1. Up to row 56, the Introduction is rather unfocused for the topic of the article. The introductory paragraph has been shortened accordingly to highlight the role of TGFb and the remaining topics of our article
  2. 2. In the last paragraph it is inferredthat the authors did tests in order to analyze the TGF-ß physiological role. We added a sentence to clarify that our work has been made on previous literature

Section 2. 

  1. This section needs rephrasing in a less convoluted manner. A red string needs to be followed in between paragraphs. The section has been rewritten as requested; first, it analyze classic TGFb pathway, then possible pathological alterations in CRC, the last part is about other non canonical pathways.
  2. Figure 1 needs to be of higher quality. Please make sure you have the consent of the original authors for any figure you are borrowing from previously published material. Image has been created by the authors for the article. We increased DPI to 300 dpi.

Section 3. 

  1. Please use full name before using abbreviations (ZO – line 130) Corrected
  2. Please make sure you have the consent of the original authors for any figure you are borrowing from previously published material. Image has been created by the authors for the article.
  3. This section needs rephrasing in a less convoluted manner. A red string needs to be followed in between paragraphs. The section has been rephrased as requested and several sentences shortened to avoid redundancy. There is first an introduction on EMT, then TGFb role with both SMAD dependent and SMAD independent pathways

Section 5. 

  1. TME is an abbreviation not explained before its usage. Corrected
  2. Please make sure you have the consent of the original authors for any figure you are borrowing from previously published material. Image has been created by the authors for the article
  3. Figure 4 has a missing arrowhead in the positive feedback loop involving the RUNX3/MYC/TGF-ß pathway. Image has been corrected

Sections 6,7&8

  1. Please try to follow a line of argumentation with the information provided.
    Section 6 has been slightly rephrased: as per the section title, the focus is on additional but indipendent roles of TGFb in carcinogenesis. Section 7 has been slightly remodeled: we first analysed clinical trials with anti TGFb and their shortcoming, then we focused on other approaches as TKIs, LncRNAs and Sitagliptin; in the last paragraph, we describe an unwanted effect of TGFb inhibition that may limit our expectations on TGFb usefulness as a therapeutic target. Section 8 has been modified too, with addition of some details and a conclusion to the role of TGFb in restoring chemosensitivity
  2. Please summarize each section into a figure, as provided for previous sections. This would rapidly increase readers’ understanding. Figure 5 and 6 have been added. A table, in lieu of a figure, has been added for the Treatment section, as suggested by Reviewer 2

OVERALL REMARKS: 

A proofreading of the manuscript is necessary. There are spelling, punctuation, and formatting mistakes throughout the entire manuscript. Please organize the information provided in the review so that the reader can get the most important parts in the most optimal manner. We revised the manuscript accordingly

Round 2

Reviewer 1 Report

Comments and Suggestions for Authors

As I already mentioned in my previous reports, the topic of this review is widely addressed in other work and the manuscript in itself does not provide the sufficient degree of detail to give a significant contribution to the field. This means that the manuscript needed at least a substantial revision, a task that the authors did not accomplish, since they just added some short sections that do not amend the overall flaws of the manuscript. In order to help the authors to understand the weaknessess of their manuscript, below I provide a detailed list of the many aspects that should be improved:

- lines 91-105: Mutations in TGF-β receptors and SMAD proteins frequently occur in CRC. Which kind of mutation? Gain/loss of function? How do they affect CRC development? Are there differences in mutations type/amount between sporadic and familiar CRC? Between in situ CRC and mCRC?

- lines 106-109: how do TGFb signaling activate these non-canonical pathways? Paragraph 2 is introductive; it would be more important to explain how these pathways work than mentioning their effects on other processes that are discussed later.

- lines 114-131: this section is full of repetitions from one side and poor in details from the other. The authors repeat that a series of signaling events causes the expression of mesenchymal proteins, the loss of cell polarity and the acquisition of a mobile and invasive phenotype. This could be said once. Intesead, they do not mention the signaling events involved. What are they? How does loss of E-cadherin occur? How does it act on cytoskeleton? Does it affect the expression of mesenchymal proteins? How?

Lines 132-142: the authors mention that “TGF-β may activate a number of cellular signals and molecular mechanisms contributing to tumor progression, promoting EMT and cancer stem cells (CSCs) proliferation, but actually describe just one mechanism mediated by SMAD4. Which are the others? How does TGFb act on CSC, since it is not described at all? How does the induction of Snail-1, Slug and Twist-1 repress E-cadherin? Are these the only genes involved in EMT and regulated by SMAD4 or there are others?

- Lines 143-150: the authors say that SMAD-independent TGFb signaling induces cytoskeleton dissolution and ERK activation. How does it occur? What is the molecular mechanism linking TFGb to cytoskeleton and ERK activation?

- Lines 157-172: how does TGF-b regulate the expression of cited miRNAs and lcRNA? Through SMADs or other mechanisms? How do TUG1 and LINC01133 affect EMT?

- Lines 176-181: how does this phenomenon occur? How does the mutation modify the activity of the receptor?

- Lines 182-190: This section is completely unclear. Based on the authors say, HIF-1a appears to up-regulate both E-cadherin and mesenchymal proteins? Thus, why the net effect is EMT induction? Moreover, how does HIF-1a up-regulate TGF-b? How does then TGF-b interact with HIF-1a?

Lines 202-215: Do the mentioned phenomena depend on SMAD4 or not? What are the other signaling pathways engaged by TGF-b to induce these effects?

Lines 222-224: this sentence is a mix of notions without a clear sense. Form the sentence, it seems that the balance between VEGF and TGF-b signaling is needed for vessels organization in physiological conditions. When it is dysregulated in can induce tumor growth. However, it is not just a question of VEGF signaling overcoming TGF-b one, since, as the authors mention, TGF-b signaling contribute to enhance VEGF pathway. The authors should rewrite this sentence more consistently with the rest of the paragraph.

- Lines 232-236: how does this exhaustion occur? There is the up-regulation of inhibitory receptors that could justify the effectiveness of combining TGF-b inhibitors with immune checkpoints inhibitors?

- Lines 238-244: this section is poorly written and misleading. Thea authors should firstly mention the general mechanism by which THSD4 repress TGF-b signaling, and then describe how over-expression of ZNF37A repress THSD4, allowing TGF-b to induce fibroblasts conversion into CAFs. Moreover, the authors should clearly indicate that TGF-b is produced by tumor cells

- Lines 273: how does CTHCR1 activate TGF-b signaling?

- Lines 290-294: it is not specified whether the up-regulation of TGF-b and PODXL are linked or just occur together. The authors should better clarify this point. In case the cause-effect link has not been demonstrated, the authors should mention that the evidence is just suggestive of a role for TGFb in mediating ECM deposition.

- Lines 301-304: there are hypotheses about how inactivation of TGF-bR2 promotes vascular invasion? If yes, they should be mentioned; otherwise, it should be said that the underlying mechanisms of this association are not known

- Lines 359-364: the mentioned inhibitors are indicated as kinase inhibitors. Are they specific for TGFb receptor or there can be off-target effects? If yes, is it possible that the positive effects could be mediated by a more general inhibition of pro-tumoral kinases activities?

- Lines 409-415: this section needs more details. What is the connection between TGF-b, p53 and resistance?

- Lines 416-422: what is the possible mechanism by which BRAFi up-regulate TGF-b signaling? At which level?

- Lines 425-428: this section poorly fits with what the authors say in the previous section. In theory, TGF-b represent a useful target for therapy, but evidences from clinical trial indicate that the current approaches are not feasible due to poor effectiveness and high toxicity. Thus, how do the authors think that TGF-b inhibition could be associated with chemotherapy? Which strategies they propose? What are the aspects that should be addressed in order to effectively use TGF-b inhibition in clinical setting?

- Lines 432-437: the conclusion does not completely recapitulate the content of the review. The lack of effectiveness observed by blocking TGF-b signaling should be highlighted, and a brief description of how to improve these disappointing results should be added

Comments on the Quality of English Language

Minor revisions are needed. Please accurately check the manuscript and use an uniform nomeclature for TGF-b

Author Response

We tried to answer all queries by the reviewer at the best of our possibilities; unfortunately, lines as reported in the review and lines in the final manuscript did not match, we hope it did not influence much our work.

- lines 91-105: Mutations in TGF-β receptors and SMAD proteins frequently occur in CRC. Which kind of mutation? Gain/loss of function? How do they affect CRC development? Are there differences in mutations type/amount between sporadic and familiar CRC? Between in situ CRC and mCRC? Details have been added.

- lines 106-109: how do TGFb signaling activate these non-canonical pathways? Paragraph 2 is introductive; it would be more important to explain how these pathways work than mentioning their effects on other processes that are discussed later. Details have been added.

- lines 114-131: this section is full of repetitions from one side and poor in details from the other. The authors repeat that a series of signaling events causes the expression of mesenchymal proteins, the loss of cell polarity and the acquisition of a mobile and invasive phenotype. This could be said once. Intesead, they do not mention the signaling events involved. What are they? How does loss of E-cadherin occur? How does it act on cytoskeleton? Does it affect the expression of mesenchymal proteins? How? Details have been added.

Lines 132-142: the authors mention that “TGF-β may activate a number of cellular signals and molecular mechanisms contributing to tumor progression, promoting EMT and cancer stem cells (CSCs) proliferation, but actually describe just one mechanism mediated by SMAD4. Which are the others? SMAD independent mechanisms are already explained in the text [On the other hand, Smad-independent activity promotes dissolution of cell junction complexes etc] How does TGFb act on CSC, since it is not described at all? Details have been added. How does the induction of Snail-1, Slug and Twist-1 repress E-cadherin?  Details have been added. Are these the only genes involved in EMT and regulated by SMAD4 or there are others? Details have been added.

- Lines 143-150: the authors say that SMAD-independent TGFb signaling induces cytoskeleton dissolution and ERK activation. How does it occur? What is the molecular mechanism linking TFGb to cytoskeleton and ERK activation? Details have been added.

- Lines 157-172: how does TGF-b regulate the expression of cited miRNAs and lcRNA? Through SMADs or other mechanisms? How do TUG1 and LINC01133 affect EMT? Details have been added

- Lines 176-181: how does this phenomenon occur? How does the mutation modify the activity of the receptor? Details and a new reference have been added

- Lines 182-190: This section is completely unclear. Based on the authors say, HIF-1a appears to up-regulate both E-cadherin and mesenchymal proteins? Thus, why the net effect is EMT induction? Moreover, how does HIF-1a up-regulate TGF-b? How does then TGF-b interact with HIF-1a? Details have been added. E-Cadherin was an error.

Lines 202-215: Do the mentioned phenomena depend on SMAD4 or not? What are the other signaling pathways engaged by TGF-b to induce these effects? Details have been added

Lines 222-224: this sentence is a mix of notions without a clear sense. Form the sentence, it seems that the balance between VEGF and TGF-b signaling is needed for vessels organization in physiological conditions. When it is dysregulated in can induce tumor growth. However, it is not just a question of VEGF signaling overcoming TGF-b one, since, as the authors mention, TGF-b signaling contribute to enhance VEGF pathway. The authors should rewrite this sentence more consistently with the rest of the paragraph. As the sentence was unclear and thorough discussion could be too long for the scope of the paragraph, the sentence has been deleted.

- Lines 232-236: how does this exhaustion occur? There is the up-regulation of inhibitory receptors that could justify the effectiveness of combining TGF-b inhibitors with immune checkpoints inhibitors? Details have been added.

- Lines 238-244: this section is poorly written and misleading. Thea authors should firstly mention the general mechanism by which THSD4 repress TGF-b signaling, and then describe how over-expression of ZNF37A repress THSD4, allowing TGF-b to induce fibroblasts conversion into CAFs. Moreover, the authors should clearly indicate that TGF-b is produced by tumor cells.
Section has been rephrased.

- Lines 273: how does CTHCR1 activate TGF-b signaling? Details have been added.

- Lines 290-294: it is not specified whether the up-regulation of TGF-b and PODXL are linked or just occur together. The authors should better clarify this point. In case the cause-effect link has not been demonstrated, the authors should mention that the evidence is just suggestive of a role for TGFb in mediating ECM deposition. Details have been added.

- Lines 301-304: there are hypotheses about how inactivation of TGF-bR2 promotes vascular invasion? If yes, they should be mentioned; otherwise, it should be said that the underlying mechanisms of this association are not known. Details have been added.

- Lines 359-364: the mentioned inhibitors are indicated as kinase inhibitors. Are they specific for TGFb receptor or there can be off-target effects? If yes, is it possible that the positive effects could be mediated by a more general inhibition of pro-tumoral kinases activities? No off-targets effects have been reported in literature.

- Lines 409-415: this section needs more details. What is the connection between TGF-b, p53 and resistance? Details have been added.

- Lines 416-422: what is the possible mechanism by which BRAFi up-regulate TGF-b signaling? At which level? Results from the study by Sun et al have been added

- Lines 425-428: this section poorly fits with what the authors say in the previous section. In theory, TGF-b represent a useful target for therapy, but evidences from clinical trial indicate that the current approaches are not feasible due to poor effectiveness and high toxicity. Thus, how do the authors think that TGF-b inhibition could be associated with chemotherapy? Which strategies they propose? What are the aspects that should be addressed in order to effectively use TGF-b inhibition in clinical setting? We slightly rephrased the section, further clarifying that that, while addition of TGFb inhibitor may be useful in clinical practice both because TGFB appear to be a useful target and because it may help in reduce chemoresistance, at the moment no TGFb inhibitor is able to overcome the limitations exposed in the previous paragraph. All discussion relative to use of TGFb inhibitor in a clinical setting is accordingly in the previous paragraph.

- Lines 432-437: the conclusion does not completely recapitulate the content of the review. The lack of effectiveness observed by blocking TGF-b signaling should be highlighted, and a brief description of how to improve these disappointing results should be added. Conclusion paragraph has been rewritten.

Round 3

Reviewer 1 Report

Comments and Suggestions for Authors

There are some parts that still need to be improved:

Lines 102-104: it should be specified the functional effect(s) of the cited mutations.

Lines 104-106: these two sentences could be better linked

Lines 117-122: this section needs more details. Which cell cycle events are specifically regulated by RAS/TGF-b? What do the authors mean with "autoinduction" of TGF-b RI?

Lines 139-141: how does E-cadherin loss induce the mentioned phenomena? What are the molecular mechanisms?

Lines 147-150: the mentioned mechanisms do not occur in CRC. There are some hints that similar processes could occur also in CRC? Otherwise, the whole sentence is misleading, and the authors should clarly specify the systems in which observations have been made

Lines 211-220: I recommend the authors to use a homogeneous nomenclature for molecules. Here, ALK5 could not be immediately associated to TGF-b RI by the reader. Moreover, the reference refers to renal carcinoma and not CRC. The authors should clearly mention that these evidences have not been observed in CRC ad thus the possibility that this mechanism also occurs in CRC is just speculative

Lines 343-345: this sentence is not clear

Lines 470-474: this section is unclear and needs to be rewritten

Comments on the Quality of English Language

I also recommend the authors to carefully check their nomenclature and acronyms and to carefully read the manuscript in order to better blend the new parts with the previous ones. Also, please put attention to verbs and singular/plural forms, since I detected several little mistake

Author Response

  • Lines 102-104: it should be specified the functional effect(s) of the cited mutations.
  • Lines 104-106: these two sentences could be better linked. CORRECTED.
  • Lines 117-122: this section needs more details. Which cell cycle events are specifically regulated by RAS/TGF-b? What do the authors mean with "autoinduction" of TGF-b RI? CORRECTED
  • Lines 139-141: how does E-cadherin loss induce the mentioned phenomena? What are the molecular mechanisms? Sentence has been slightly rewritten.
  • Lines 147-150: the mentioned mechanisms do not occur in CRC. There are some hints that similar processes could occur also in CRC? Otherwise, the whole sentence is misleading, and the authors should clarly specify the systems in which observations have been made. A premise has been added to further clarify that these phenomena have not been yet demonstrated in CRC.
  • Lines 211-220: I recommend the authors to use a homogeneous nomenclature for molecules. Here, ALK5 could not be immediately associated to TGF-b RI by the reader. Moreover, the reference refers to renal carcinoma and not CRC. The authors should clearly mention that these evidences have not been observed in CRC ad thus the possibility that this mechanism also occurs in CRC is just speculative
  • Lines 343-345: this sentence is not clear CORRECTED.
  • Lines 470-474: this section is unclear and needs to be rewritten. CORRECTED.
  • Comments on the Quality of English Language: I also recommend the authors to carefully check their nomenclature and acronyms and to carefully read the manuscript in order to better blend the new parts with the previous ones. Also, please put attention to verbs and singular/plural forms, since I detected several little mistake. CORRECTED